# Serum N-terminal telopeptide of type I collagen as a biomarker for predicting bone density loss in patients with Crohn disease

**Natsuki Ishida[1], Tomohiro Higuchi[1], Takahiro Miyazu[1], Satoshi Tamura[1], Satoshi Suzuki[2], Shinya Tani[1], Mihoko Yamade[1], Moriya Iwaizumi[3], Yasushi Hamaya[1], Satoshi Osawa[2], Takahisa Furuta[4], Ken Sugimoto[1]\***

1 First Department of Medicine, Hamamatsu University School of Medicine, Hamamatsu, Shizuoka, Japan, 2 Department of Endoscopic and Photodynamic Medicine, Hamamatsu University School of Medicine, Hamamatsu, Shizuoka, Japan, 3 Department of Laboratory Medicine, Hamamatsu University School of Medicine, Hamamatsu, Shizuoka, Japan, 4 Center for Clinical Research, Hamamatsu University School of Medicine, Hamamatsu, Shizuoka, Japan

\* sugimken@hama-med.ac.jp

**Data Availability Statement:** All data needed to evaluate and reach the conclusions in the paper are presented in this paper. Additional data related to this study may be requested from the authors.

## Abstract

### Background

The serum N-terminal telopeptide of type I collagen (NTx) is significantly higher in patients with Crohn disease (CD) than in healthy individuals and patients with ulcerative colitis. This study aimed to investigate whether an elevated serum NTx level is a risk predictor of osteoporosis in patients with CD.

### Methods

Based on whether the femoral Z-score decreased over a 2-year period, 41 CD patients were divided into the ΔZ-score <0 group (Z-score decreased) and the ΔZ-score ≥0 group (Z-score did not decrease). The risk predictors of a femoral Z-score decrease were examined. Furthermore, we investigated the correlations between the ΔZ-score (which represents the change in the Z-score over a 2-year period) and the mean levels of biomarkers, including the Crohn Disease Activity Index (CDAI), serum albumin, C-reactive protein, and bone metabolism markers (including NTx) measured initially (i.e., in our previous study) and 2 years later (present study). The relationships between anti-tumor necrosis factor α (anti-TNF-α) therapy and serum NTx levels were also examined.

### Results

Although there was no correlation between the mean CDAI and the ΔZ-score, the mean serum NTx and albumin levels were significantly correlated with the ΔZ-score (P<0.01 and P = 0.02, respectively). Furthermore, the femoral Z-score tended to be lower in the anti-TNF-α administration group than in the non-administration group.

**Funding:** The authors received no specific funding for this work.

**Competing interests:** The authors have declared that no competing interests exist.

## Conclusions

These observations indicated that an elevated serum NTx could be a useful marker for predicting a decrease in the femoral bone mineral density in CD patients. Anti-TNF-α therapy maintained an elevated serum NTx level, suggesting that treatment with anti-TNF-α may help control increased bone resorption in CD patients.

## Introduction

Inflammatory bowel disease (IBD), which includes Crohn disease (CD) and ulcerative colitis (UC), is associated with a high incidence of osteoporosis [1–6]. Approximately 18%–42% and 22%–77% of the patients with IBD have osteoporosis and reduced bone mineral density (BMD), respectively [7, 8]. In fact, the risk of fracture is high in IBD patients, and the relative risk of all fractures is reported to be 1.41 (95% confidence interval [CI]: 1.27–1.56, P<0.001) [9]. Because osteoporosis carries a fracture risk, prevention of osteoporosis in IBD is important because fractures significantly reduce the patient's quality of life.

Osteoporosis is diagnosed by measuring the BMD using dual-energy X-ray absorptiometry (DEXA), the results of which are presented as the Z- and T-scores [10–13]. Furthermore, biochemical bone metabolism markers are also used for evaluating the state of bone metabolism. These markers are classified into bone formation markers, which include bone-specific alkaline phosphatase (BAP) and the N-terminal propeptide of type I collagen; bone resorption markers, which include the C- and N-terminal telopeptides (NTx) of type I collagen; and bone matrix-related markers, which include undercarboxylated osteocalcin (ucOC; related to vitamin K deficiency) [14–17].

Bone metabolism is also associated with the activity of proteases, including the matrix metalloproteinases, and with the disease activity matrix metalloproteinases in IBD [18–20]. Furthermore, IBD is associated with hypoalbuminemia due to malnutrition, which is independently associated with the risk of osteoporosis in the femoral neck, hips, and lumbar spine. A longer duration of hypoalbuminemia was reported to increase the risk of osteoporosis at the same anatomical site [21] and at different anatomical sites [22].

In a previous cross-sectional study [23], we examined the association between biochemical bone metabolism markers (namely BAP, NTx, ucOC, and 1,25-dihydroxyvitamin D [1,25-$(OH)_2D$]) and bone density in IBD patients and reported that the serum NTx level, a biochemical marker predicting an increase in bone resorption, is significantly elevated in patients with CD receiving infliximab (IFX). However, in this study, the BMD and bone metabolism markers were examined in a single measurement, and we realized the need to investigate long-term changes in bone metabolism over time. Therefore, in this study, the BMD and bone metabolism markers were measured again 2 years after the initial measurement in the previous study, and the differences between them were evaluated.

## Materials and methods

### Ethics statements

The study protocol was reviewed and approved by the Institutional Review Board of Hamamatsu University School of Medicine (16–246). The investigation was conducted in accordance with Good Clinical Practice principles in adherence to the Declaration of Helsinki. Written informed consent was obtained from all individual participants included in the study.

## Aim, design, and setting

We aimed to investigate whether elevated serum NTx level is a risk predictor of osteoporosis in patients with CD. In this prospective study, we examined the significance of various bone metabolism biomarkers, including the serum NTx levels, in CD patients who were treated at the Hamamatsu University School of Medicine between July 2013 and February 2016 who were having a tendency for reduced BMD.

## Patients

A total of 41 patients (34 male and 7 female) with CD were enrolled in this study. We selected CD patients as the study subjects because vitamin and mineral absorption disorders are likely to lower the BMD in these patients [24, 25]. The exclusion criteria were refusal to provide informed consent, pregnancy or a desire to become pregnant (to protect such women from the effects of radiation exposure during DEXA scanning), cancers (especially bone cancers), and UC, Behçet disease, and other IBD such as indeterminate colitis. Patients' data such as age, disease duration and location, operation history, and treatment were recorded.

## Biochemical measurements

Because bone metabolism markers are easily affected by the diet, serum samples for biochemical measurements were collected under fasting conditions [26]. This method has been described previously [23]. Briefly, the blood samples were centrifuged at 3,000 rpm for 10 min at 4 ˚C. The serum samples (top layer) were separated and stored at −20 ˚C until analysis. The serum levels of bone metabolism markers, namely ucOC, 1,25-(OH)$_2$D, NTx, and BAP, were estimated by an electrochemiluminescence immunoassay (Sanko, Tokyo, Japan), radioimmunoassay (Immunodiagnostic Systems, Boldon, UK), enzyme-linked immunosorbent assay (Ostex, Seattle, WA, USA), and enzyme immunoassay (Quidel, San Diego, CA, USA), respectively. All laboratory measurements were blinded to the investigators and were performed at a Special Research Laboratory inc., an outside contractor [23].

## Disease assessment

CD activity was evaluated using the CD Activity Index (CDAI) [27]. CDAIs between 150 and 220, 220 and 450, and ≥450 indicated mild, moderate, and severe activity, respectively. The remission maintenance rate was evaluated with 150 as the cutoff CDAI. Serum C-reactive protein (CRP) level and serum albumin were measured for the assessment of CD activity and the patient's nutritional status, respectively. These measurements were performed at the Laboratory Test Department of the Hamamatsu University School of Medicine.

## Measurement of BMD and determination of the Z-scores

This method has been described previously [23]. Briefly, the BMD of the L2–L4 lumbar vertebrae was measured initially and 2 years later using the Discovery DXA System (Discovery A; Hologic, Bedford, MA, USA) and expressed using absolute values (g/cm$^2$) and Z-scores [28]. The Z-scores were based on the standard deviation for the mean scores of a reference young adult sex-matched population. The ΔZ-score represents the changes in the Z-scores measured initially (i.e., in the previous study) and 2 years later. Accordingly, the enrolled CD patients were divided into two groups, namely, one in which the femoral Z-score decreased (i.e., the ΔZ-score <0 group; n = 20) and one in which the femoral Z-score did not decrease (i.e., the ΔZ-score ≥0 group; n = 21) 2 years after the initial measurement. The patients' characteristics were compared between the two groups. Furthermore, the relationship among the clinical

score (CDAI), laboratory test values (CRP and albumin), and bone metabolism markers (1,25-(OH)$_2$D, BAP, NTx, and ucOC) was investigated in these two groups.

## Statistical analysis

All statistical analyses were performed using SPSS version 24 (IBM Corp., Armonk, NY, USA) and SAS version 9.4 (SAS Institute, Cary, NC, USA). Data are expressed as mean±standard deviation. The chi-square test and Fisher exact test were used to compare the test variables between the ΔZ-score <0 and ΔZ-score ≥0 groups. The correlations between two independent measurements were assessed using the Pearson's correlation coefficient: We examined the correlation between the femoral ΔZ-score and the mean value of each biomarker, i.e., the CDAI (clinical score), serum albumin and CRP levels (laboratory test values), and bone metabolism markers estimated in the initial study and 2 years later. Furthermore, a multivariate regression model, including NTx and albumin, was constructed with the femoral neck ΔZ-score as the predictor, and a multiple regression analysis was performed with a significance level of 5%. The optimal NTx cut-off values for predicting Z score decrease were analyzed using receiver operating characteristic (ROC) analysis. The accuracy of the predictive values was evaluated using the area under the ROC curve (AUC). Logistic regression analysis was used to evaluate the odds ratio of Z score decrease during 2 years. P <0.05 was considered statistically significant. The required sample size was 16 under the condition that the difference in the mean value of NTX between the bone mineral density reduced group and the non-decreased group was "3", the standard deviation was "3", the α error (significance level) was "5% on both sides", and the detection power was "80%". Therefore, it was considered that 21 subjects in the ΔZ-score <0 group and 20 subjects in the ΔZ-score ≥0 group had a sufficient number of samples for analysis.

## Results

### Patient characteristics

The patients' baseline characteristics are shown in Table 1.

### Disease assessment

Among the 41 patients enrolled, 13 were undernourished with hypoalbuminemia: 9 (22.0%) presented with albumin levels <4.0 g/dL, whereas 4 (9.8%) presented with ≤3.5 g/dL. There was no significant correlation between the CDAI at the beginning of the present study and the bone density of the lumbar spine (P = 0.5045) or the femoral Z-score (P = 0.3947). Furthermore, there were no significant correlations between the biomarkers and the IBD phenotype.

### Comparison between the ΔZ-score <0 and ΔZ-score ≥0 groups

No significant differences in the baseline patient characteristics were observed between the two groups (Table 2). Table 3 shows a comparison of the biomarker values between the ΔZ-score <0 and ΔZ-score ≥0 groups at the initial measurement in the previous study. The mean NTx value at the first measurement was significantly higher in the ΔZ-score <0 group than in the ΔZ-score ≥0 group (Table 3); however, no significant differences in the CDAI and other biomarkers levels were observed between the two groups at the first measurement; although the 1,25-(OH)$_2$D tended to be lower in the ΔZ-score <0 group, the difference was not significant. The cut-off value of NTx at initial measurement for predicting the progression of osteoporosis (ΔZ score <0 group) in two years was 16.6 nmol BCE/L by ROC analysis, and the AUC was 0.826 (95% confidence interval (CI) 0.694–0.958) (Fig 1). Furthermore, univariate

**Table 1. Baseline characteristics of patients with Crohn disease.**

| Patient characteristics | N = 41 |
| --- | --- |
| Age at initial measurement (years), mean±SD (range) | 38.4±9.9 (20–58) |
| Male/female patients, n (%) | 34 (82.9)/7 (17.1) |
| Age at diagnosis (years), mean±SD (range) | 27.3±11.1 (13–58) |
| Disease duration (years), mean±SD (range) | 11.1±7.9 (0–29) |
| Smoker/non-smoker, n (%) | 33/8 (80.5/19.5) |
| Age at diagnosis (A1, younger than 16 years of age; A2, between 17 and 40 years of age; A3, older than 40 years of age) | |
| A1/A2/A3, n (%) | 5 (12.2)/30 (73.2)/6 (14.6) |
| Current disease location (L1, ileal; L2, colonic; L3, ileocolonic; L4, isolated upper disease) | |
| L1/L2/L3/L4, n (%) | 16 (39.0)/3 (7.3)/22 (53.7)/0 (0) |
| Current disease behavior (B1, non-stricturing, non-penetrating; B2, stricturing; B3, penetrating) | |
| B1/B2/B3 | 12 (29.3)/14 (34.1)/15 (36.6) |
| B1/B2+B3, n (%) | 12 (29.3)/29 (70.7) |
| Active perianal fistula, n (%) | 8 (19.5) |
| Previous surgery (%), n (%) | 18 (43.9) |
| Treatment at initial measurement, n (%) | |
| 5-Aminosalicylate | 37 (90.2) |
| Prednisolone | 0 (0) |
| Immunomodulators | 15 (36.6) |
| Anti-TNF-α therapy | 25 (61.0) |
| Enteral nutrition | 20 (48.8) |

SD, standard deviation; TNF, tumor necrosis factor.

and multivariate analysis showed that NTx $\geq$ 16.6 nmol BCE/L was an independent risk factor with osteoporosis progression ($\Delta$Z score <0) (Table 4).

Table 5 shows the average clinical scores, laboratory test values, and bone metabolism marker levels of the first and second measurements. There were no significant differences in the serum CRP level between the $\Delta$Z-score <0 and $\Delta$Z-score $\geq$0 groups. The average serum albumin level was significantly lower in the $\Delta$Z-score <0 group than in the $\Delta$Z-score $\geq$0 group at both the first and second measurements. As observed during the first measurement, the mean serum NTx level during the second measurement was significantly higher in the $\Delta$Z-score <0 group than in the $\Delta$Z-score $\geq$0 group. Although NTx was not a predictor of the femoral neck $\Delta$Z-score when the significance level was set to 5%, it was an independent predictor when the significance level was set to 10%.

We then evaluated the changes in the values of each biomarker at initial and second measurements in the group with $\Delta$Z score <0 and the group with $\Delta$Z score $\geq$ 0 (Table 6). Alb showed a significant decrease in the group with $\Delta$Z score <0 (P = 0.02). Interestingly, NTx did not change significantly in the group with $\Delta$Z score <0, whereas the value of NTx decreased significantly in the group with $\Delta$Z score $\geq$ 0 (P = 0.92 and P <0.01, respectively).

## Correlation of the $\Delta$Z-score with the CDAI, serum NTx level, and serum albumin

The correlations between the $\Delta$Z-score and the average CDAI, serum NTx, and serum albumin levels at the first and second measurements were examined. No significant correlation was found between the mean CDAI and the femoral $\Delta$Z-score (r = 0.1701, P = 0.2878) (Fig 2a). However, a

**Table 2. Comparison of patient characteristics between those with an ΔZ-score <0 and those with an ΔZ-score ≥0.**

| Patient characteristics | ΔZ-score <0, N = 21 | ΔZ-score ≥0, N = 20 | P-value |
|---|---|---|---|
| Age at initial measurement (years), mean±SD (range) | 38.3 ± 11.4 (20–58) | 38.5 ± 8.4 (22–55) | 0.96 |
| Male/female patients, n (%) | 18/3 | 16/4 | 0.63 |
| Age at diagnosis (years), mean±SD (range) | 26.9 ± 12.3 (13–58) | 27.8 ± 10.0 (13–47) | 0.81 |
| Disease duration (years), mean±SD (range) | 11.4 ± 9.3 (0–29) | 10.8 ± 6.5 (1–26) | 0.79 |
| Smoker/non-smoker, n (%) | 10 (47.6)/11 (52.4) | 8 (40.0)/12 (60.0) | 0.62 |
| Age at diagnosis (A1, younger than 16 years of age; A2, between 17 and 40 years of age; A3, older than 40 years of age) | | | |
| A1/A2/A3 | 2 (9.5)/16 (76.2)/3 (14.3) | 3 (15.0)/14 (70.0)/3 (15.0) | 0.86 |
| Current disease location (L1, ileal; L2, colonic; L3, ileocolonic; L4, isolated upper disease) | | | |
| L1/L2/L3 | 8 (38.1)/1 (4.8)/12 (57.1) | 8 (40.0)/2 (10.0)/10 (50.0) | 0.78 |
| Current disease behavior (B1, non-stricturing, non-penetrating; B2, stricturing; B3, penetrating) | | | |
| B1/B2/B3 | 6 (28.6)/8 (38.1)/7 (33.3) 6 | 6 (30.0)/6 (30.0)/8 (40.0) | 0.85 |
| B1/B2+B3 | (28.6)/15 (71.4) | 6 (30.0)/14 (70.0) | 0.92 |
| Active perianal fistula, n (%) | 3 (14.3) | 5 (25.0) | 0.39 |
| Previous surgery (%), n (%) | 11 (52.4) | 7 (35.0) | 0.26 |
| Treatment, n (%) | | | |
| 5-Aminosalicylate | 18 (85.7) | 19 (95.0) | 0.32 |
| Prednisolone | 0 (0.0) | 0 (0.0) | - |
| Immunomodulators | 6 (28.6) | 9 (45.0) | 0.27 |
| Anti-TNF-α therapy | 15 (71.4) | 10 (50.0) | 0.16 |
| Enteral nutrition | 9 (42.9) | 11 (55.0) | 0.44 |

SD, standard deviation; TNF, tumor necrosis factor.

significantly negative correlation was observed between the mean serum NTx value and the femoral ΔZ-score (r = 0.4180, P = 0.0068) (Fig 2b). A significantly positive correlation was noted between the mean serum albumin value and the femoral ΔZ-score (Fig 2c). Although the mean NTx value was not correlated with the CDAI (P = 0.3549), there was a significant correlation between the mean serum albumin value and CDAI (P = 0.0042, r = 0.4377).

## Relationship between anti-TNF-α therapy and serum NTx value

We evaluated the relationship between anti-TNF-α therapy and the NTx value. In the anti-TNF-αadministration group, the NTx value did not differ significantly between the first and

**Table 3. Comparison of the biomarker values at the initial measurement between the groups of patients with ΔZ-score <0 and ΔZ-score >0.**

| | ΔZ-score <0, N = 21 | ΔZ-score ≥0, N = 20 | P-value |
|---|---|---|---|
| CDAI, mean±SD (range) | 100.6±73.0 | 73.8±67.9 | 0.23 |
| CRP level (mg/dL), mean±SD (range) | 0.39±0.64 | 0.43±0.89 | 0.87 |
| Alb level (g/dL), mean±SD (range) | 4.0±0.6 | 4.2±0.5 | 0.30 |
| 1,25-(OH)2D level (pg/mL), mean±SD (range) | 54.0±14.9 | 57.0±15.6 | 0.53 |
| BAP level (U/L) | 14.4±4.0 | 15.1±6.9 | 0.66 |
| NTx level (nmol BCE/L), mean±SD (range) | 20.6±5.2 | 17.3±3.7 | 0.03 |
| ucOC level (ng/mL), mean±SD (range) | 7.6±3.1 | 6.6±3.1 | 0.32 |

SD, standard deviation; CDAI, Crohn Disease Activity Index; CRP, C-reactive protein; Alb, albumin; BAP, bone-specific alkaline phosphatase; NTx, N-terminal telopeptide of type I collagen; ucOC, undercarboxylated osteocalcin; 1,25-(OH)2D, 1,25-dihydroxyvitamin D.

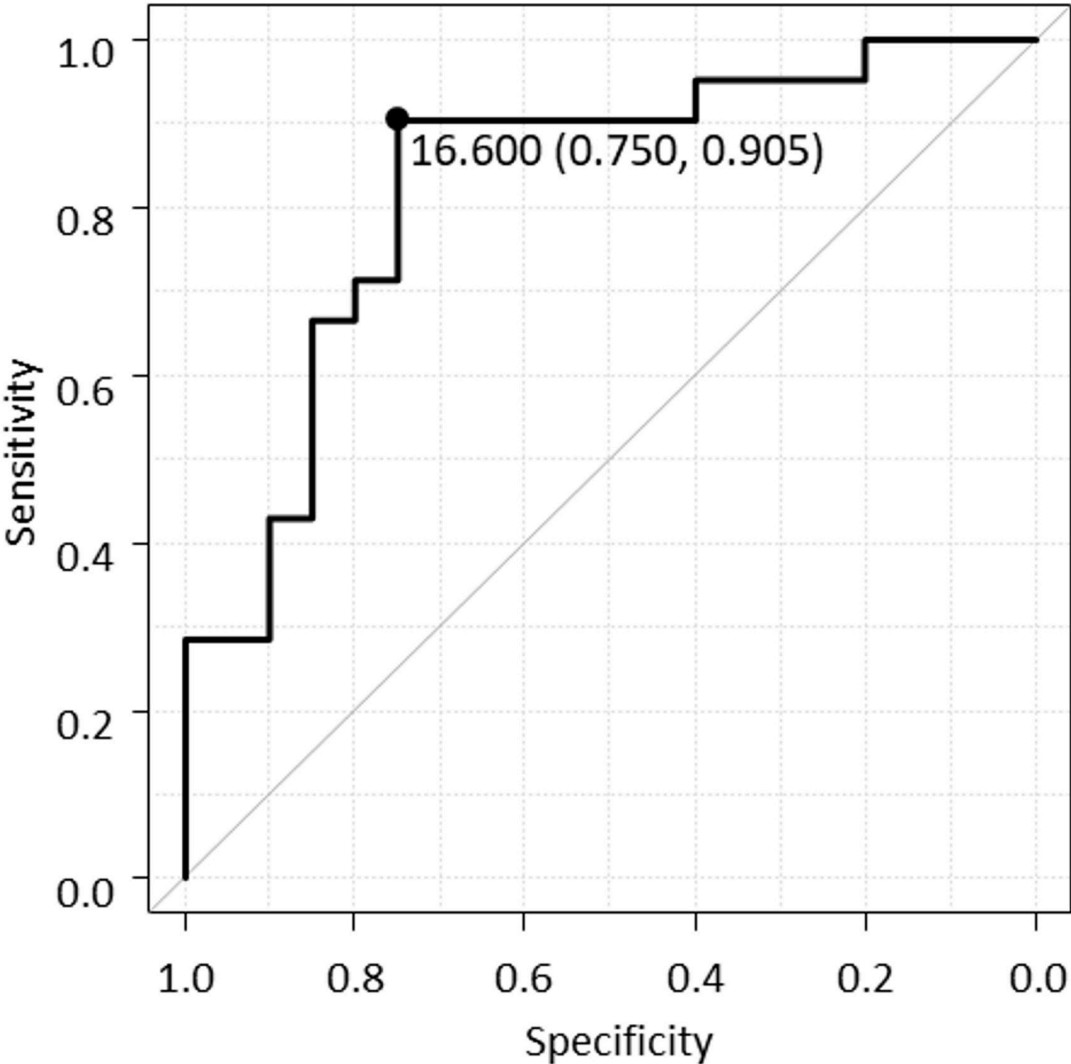

**Fig 1. Receiver operating characteristic (ROC) curve of NTx for predicting the Z score decrease for 2 years.** ROC analysis indicates an optimal cut-off value of NTx predicting Z score decrease is 16.6 nmol BCE/L, and the area under the curve is 0.826 (95% confidence interval 0.694–0.958).

second measurements (P = 0.8735) (Fig 3). However, in the anti-TNF-α non-administration group, the NTx value was significantly lower at the second measurement than at the first measurement (P = 0.0328).

## Relation between anti-TNF-α therapy and the Z-score

We compared the 2-year changes in the femoral Z-score between the anti-TNF-α administration and non-administration groups. Although the femoral neck Z-score at the first measurement tended to be lower in the anti-TNF-α administration group than in the non-administration group, this difference was not significant (P = 0.4390) (Fig 4a). Furthermore, although the femoral neck Z-score at the second measurement tended to be lower in the anti-TNF-α administration group than in the non-administration group, this was not significant either (P = 0.9219) (Fig 4b). Although the ΔZ-score tended to decrease in the anti-TNF-α

**Table 4. Multivariate analysis for predicting Z score decrease during 2 years.**

| | | Univariate analysis | | | Multivariate analysis | | |
|---|---|---|---|---|---|---|---|
| | | Odds ratio | 95%CI | P-value | Odds ratio | 95%CI | P-value |
| NTx $\geq$ 16.6 nmol BCE/L | | 28.5 | 4.830–168.0 | < 0.01 | 54.4 | 4.980–594.0 | < 0.01 |
| Age | | 1.00 | 0.937–1.06 | 0.95 | | | |
| Sex male | | 0.75 | 0.145–3.87 | 0.73 | | | |
| Age at diagnosis | A1 | 0.60 | 0.089–4.01 | 0.60 | | | |
| | A2 | 1.37 | 0.343–5.49 | 0.66 | | | |
| | A3 | 0.94 | 0.167–5.34 | 0.95 | | | |
| Current disease location | L1 | 0.92 | 0.263–3.24 | 0.90 | | | |
| | L2 | 0.45 | 0.038–5.39 | 0.53 | | | |
| | L3 | 1.33 | 0.389–4.57 | 0.65 | | | |
| Current disease behavior | B1 | 0.93 | 0.243–3.58 | 0.92 | | | |
| | B2 | 1.44 | 0.391–5.27 | 0.59 | | | |
| | B3 | 0.75 | 0.21–2.68 | 0.66 | | | |
| CDAI $\geq$ 150 | | 0.95 | 0.120–7.460 | 0.96 | | | |
| Previous surgery | | 2.04 | 0.582–7.170 | 0.26 | | | |
| Anti-TNF-α therapy | | 2.50 | 0.688–9.08 | 0.16 | | | |

95% CI, 95% confidence interval; NTx, N-terminal telopeptide of type I collagen; CDAI, Crohn Disease Activity Index; TNF, tumor necrosis factor.

administration group, there was no significant difference when compared to the score in the anti-TNF-α non-administration group during the 2-year observation period (P = 0.8556).

## Discussion

This study investigated whether an elevated serum NTx level is a risk predictor of osteoporosis in patients with CD. Results showed the BMD decreased in 20 (48.8%) of the enrolled 41 CD patients over a period of 2 years. When the CDAI, CRP and albumin levels, and bone metabolism markers were examined in the ΔZ-score <0 and ΔZ-score ≥0 groups, only the NTx value was observed to be significantly higher at the first measurement in the ΔZ-score <0 group.

The association between IBD patients and osteoporosis has been reported previously [1–8]. Furthermore, we have previously reported that compared with healthy individuals and UC patients, CD patients have a lower BMD and higher levels of bone metabolism markers [23]. It

**Table 5. Comparison of the ΔZ-score and average of the initial and secondary measured biomarkers values.**

| | ΔZ-score <0, N = 21 | ΔZ-score ≥0, N = 20 | P-value |
|---|---|---|---|
| CDAI, mean±SD | 94.0±106.6 | 64.1±63.8 | 0.13 |
| CRP (mg/dL), mean±SD | 0.49±0.84 | 0.34±0.70 | 0.37 |
| Alb (g/dL), mean±SD | 3.9±0.7 | 4.2±0.4 | 0.02 |
| 1,25-(OH)2D level (pg/mL), mean±SD | 51.3±16.4 | 56.4±14.1 | 0.13 |
| BAP level (U/L), mean±SD | 13.8±4.4 | 15.1±6.2 | 0.31 |
| NTx level (nmol BCE/L), mean±SD | 20.5±11.1 | 16.0±3.9 | 0.02 |
| ucOC level (ng/mL), mean±SD | 8.3±6.9 | 6.8±3.0 | 0.21 |

CDAI, Crohn Disease Activity Index; SD, standard deviation; CRP, C-reactive protein; Alb, albumin; BAP, bone-specific alkaline phosphatase; NTx, N-terminal telopeptide of type I collagen; ucOC, undercarboxylated osteocalcin; 1,25-(OH)2D, 1,25-dihydroxyvitamin D.
Certain data represent the mean±standard error of mean values.

**Table 6. Comparison of initial and secondary measured biomarkers values.**

| | ΔZ-score <0, N = 21 | | | ΔZ-score ≥0, N = 20 | | |
|---|---|---|---|---|---|---|
| | initial | 2 years | P-value | initial | 2 years | P-value |
| CDAI, mean±SD | 100.6±73.0 | 87.3±133.7 | 0.48 | 73.8±67.9 | 54.5±59.5 | 0.18 |
| CRP (mg/dL), mean±SD | 0.39±0.64 | 0.59±1.01 | 0.37 | 0.24±0.89 | 0.24±0.44 | 0.28 |
| Alb (g/dL), mean±SD | 4.03±0.62 | 3.81±0.83 | 0.02 | 4.22±0.48 | 4.27±0.32 | 0.47 |
| 1,25-(OH)2D level (pg/mL), mean±SD | 54.0±14.9 | 48.6±17.6 | 0.14 | 57.0±15.6 | 55.9±12.8 | 0.71 |
| BAP level (U/L), mean±SD | 14.4±4.0 | 13.4±4.7 | 0.33 | 15.1±6.9 | 15.0±5.5 | 0.86 |
| NTx level (nmol BCE/L), mean±SD | 20.6±5.2 | 20.3±15.1 | 0.92 | 17.3±3.7 | 14.7±3.7 | <0.01 |
| ucOC level (ng/mL), mean±SD | 7.62±3.10 | 8.89±9.14 | 0.36 | 6.59±3.10 | 6.95±2.98 | 0.50 |

CDAI, Crohn Disease Activity Index; SD, standard deviation; CRP, C-reactive protein; Alb, albumin; 1,25-(OH)2D, 1,25-dihydroxyvitamin D; BAP, bone-specific alkaline phosphatase; NTx, N-terminal telopeptide of type I collagen; ucOC, undercarboxylated osteocalcin.

was shown that NTx tended to be high level, especially in patients treated with infliximab. However, this study is a cross-sectional study, only comparing the values at one point in each patient, and longitudinal and temporal analysis was not performed. In the present study, we prospectively examined changes in bone metabolism markers and BMD over time and longitudinally in the same patient over 2 years. As a result, we observed that among the bone metabolism markers analyzed (BAP, NTx, and ucOC), only the bone resorption marker (NTx) was elevated; this can be explained as follows. There are two major causes of osteoporosis in CD. The first is the relationship between inflammatory cytokines and the RANK (receptor activator of NF-κβ)/RANKL (RANK ligand)/OPG system, and the second is the decreased absorption of vitamin D from the intestinal tract. Homeostasis of bone metabolism is generally maintained by the balance among RANK, bone resorption, and RANKL inhibition [29, 30]. In IBD, the activated T cells strongly express RANKL, which differentiates and induces the osteoclasts via RANK, leading to bone resorption. This mechanism decreases the BMD in IBD patients, thereby promoting bone resorption in the process. Therefore, of the three bone metabolism markers examined in this study, only the bone resorption marker, NTx, was significantly higher in the ΔZ-score <0 group than in the ΔZ-score ≥0 group at the first measurement. The mean NTx value at the first and second measurements were also significantly higher in the ΔZ-score <0 group than in the ΔZ-score ≥0 group.

Although the serum albumin level did not differ significantly between the ΔZ-score <0 and ΔZ-score ≥0 groups, the average serum albumin level at the first and second measurements in the ΔZ-score <0 group was significantly lower than in the ΔZ-score ≥0 group. This suggests that the nutritional decline in CD is related to BMD decline. Afshinnia et al. [21] reported that hypoalbuminemia is an independent risk factor for osteoporosis and that a longer duration of hypoalbuminemia carries a higher risk of osteoporosis. Our findings support that hypoalbuminemia increases the risk of osteoporosis in CD patients, as hypoalbuminemia tends to persist over time. Factors that decrease the nutritional status in CD patients include an increased metabolism due to inflammation, leakage of nutrients from the intestine, and malabsorption.

Therefore, based on the above results, we examined the correlation of serum NTx and albumin with the ΔZ-score to verify whether these are related to BMD decline over time. In this study, changes in the Z-score were shown to correlate with the serum NTx and albumin levels, indicating that these could function as predictors of BMD decline in CD patients. Conversely, there were no significant differences in the serum CRP levels between the ΔZ-score <0 and ΔZ-score ≥0 groups. Although in vitro and animal studies suggest that inflammatory cytokines affect BMD, there is insufficient evidence on the relationship between the two in CD

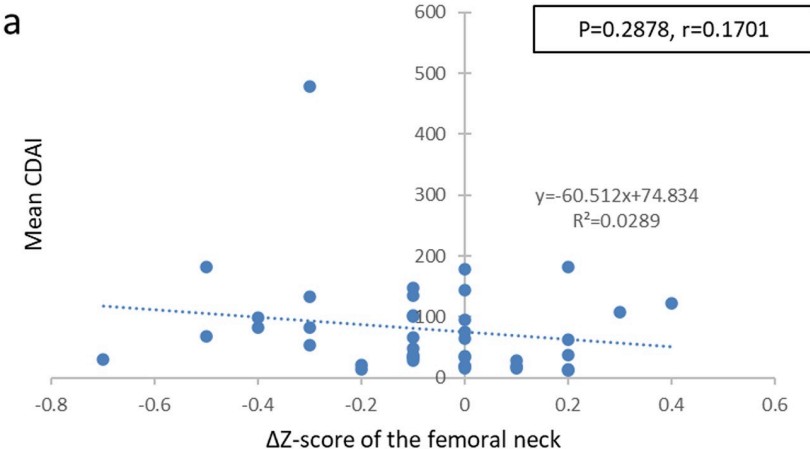

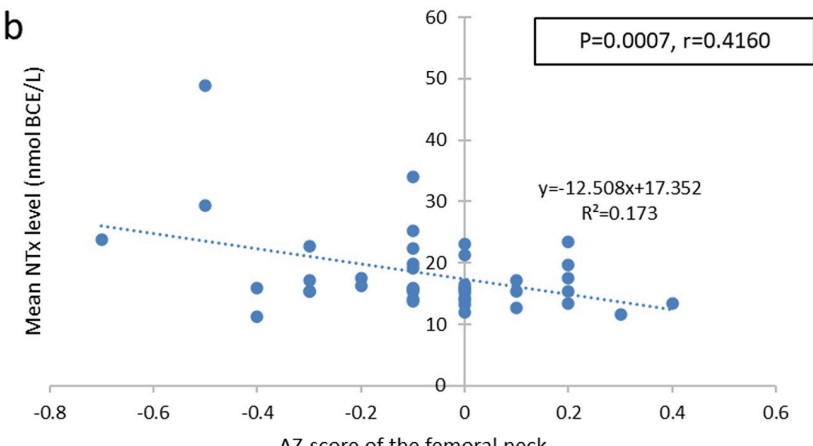

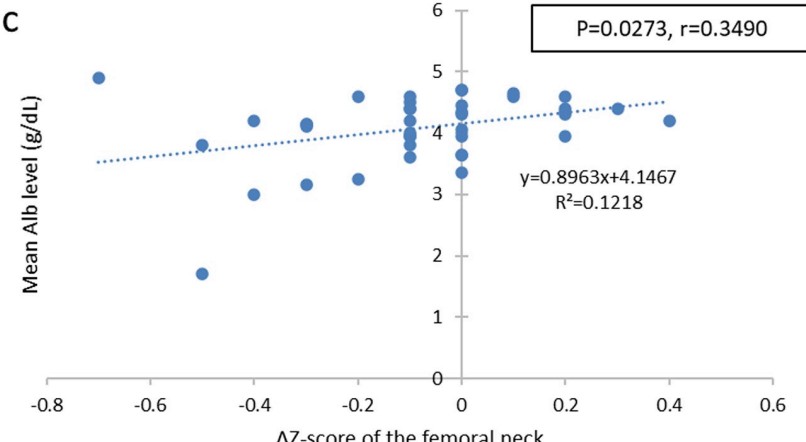

**Fig 2. Correlation between the femoral neck ΔZ-score and the mean biomarker levels.** Scatter plots of the femoral neck ΔZ-score and the mean CDAI (a), mean NTx (b), and (c) mean Alb. CDAI, Crohn Disease Activity Index; NTx, serum N-terminal telopeptide of type I collagen; Alb, albumin.

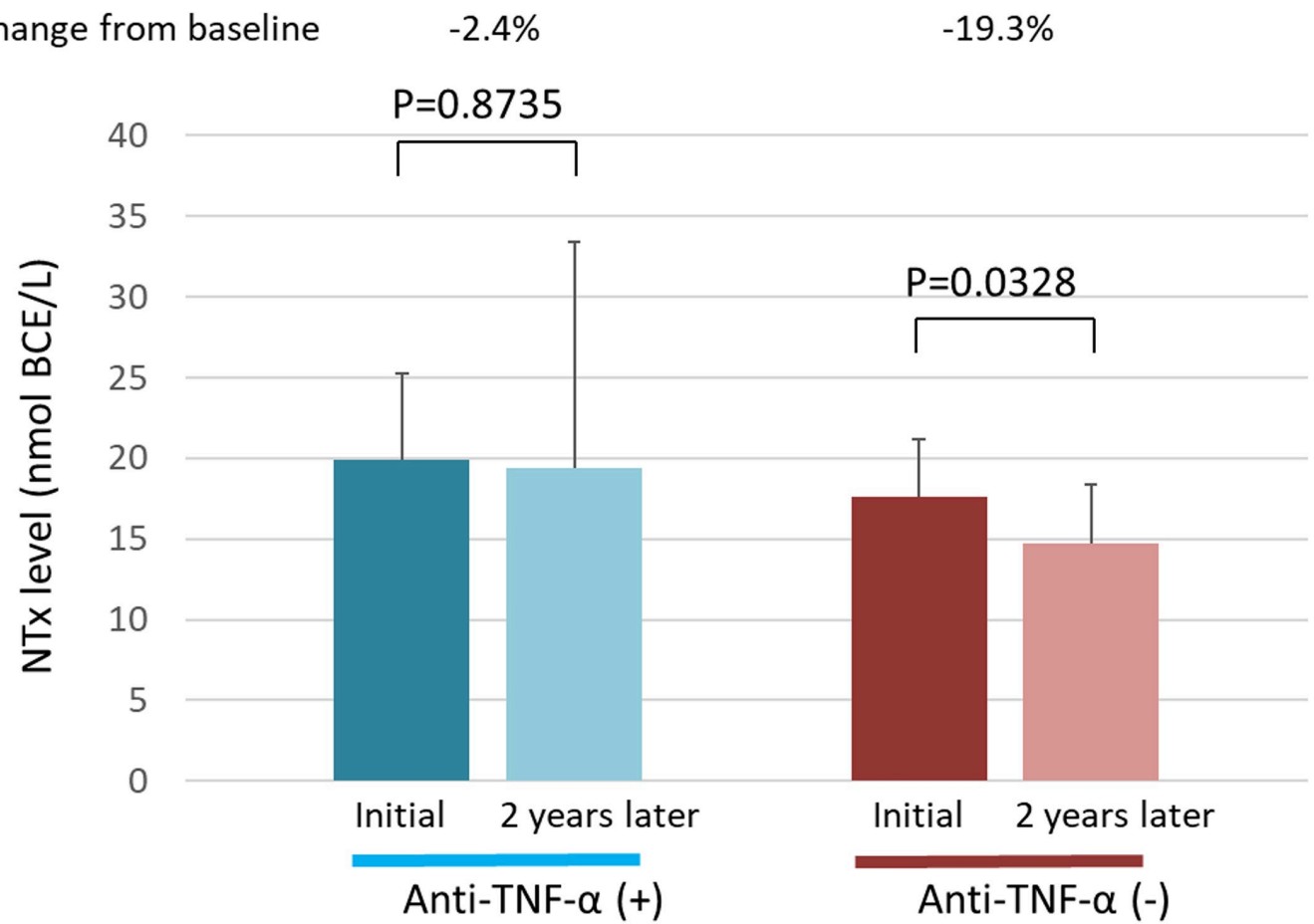

**Fig 3. Serum NTx levels in the anti-TNF-α administration and non-administration groups.** The serum NTx levels at the first measurement and 2 years later are shown for each group. NTx, N-terminal telopeptide of type I collagen; TNF-α, tumor necrosis factor α.

patients [31–33]. Although there are differences in the study conditions, some reports have indicated a relationship between the CRP level and BMD, whereas others have indicated that increased concentrations of inflammatory biomarkers are not a risk factor for BMD reduction; therefore, an effective consensus has not been achieved [34–37]. In this study, because CRP levels tended to be higher in the $\Delta$Z-score $<0$ group, we speculate that the BMD may have decreased due to an inflammatory reaction.

IBD patients, especially CD patients, are prone to malnutrition due to nutrient leakage from the intestine, small intestinal lesions, and intestinal complications. Vitamin D is required for efficient absorption of calcium from the intestinal tract, and it is well-known that patients with CD are prone to vitamin D deficiency [38, 39]. Although the 1,25-(OH)$_2$D levels did not differ significantly between the $\Delta$Z-score $<0$ and $\Delta$Z-score $\geq 0$ groups, they tended to be lower in the former.

Several reports have stated that IFX administration could improve the levels of bone metabolism biomarkers [40–42]. Veerappan et al. [43] reported that adalimumab had a beneficial effect on bone metabolism, similar to IFX. We previously showed that serum NTx levels were significantly higher in CD patients receiving IFX than in CD patients who were not [23]. In this study, we evaluated the effects of an anti-TNF-α therapy (involving not only IFX but also adalimumab

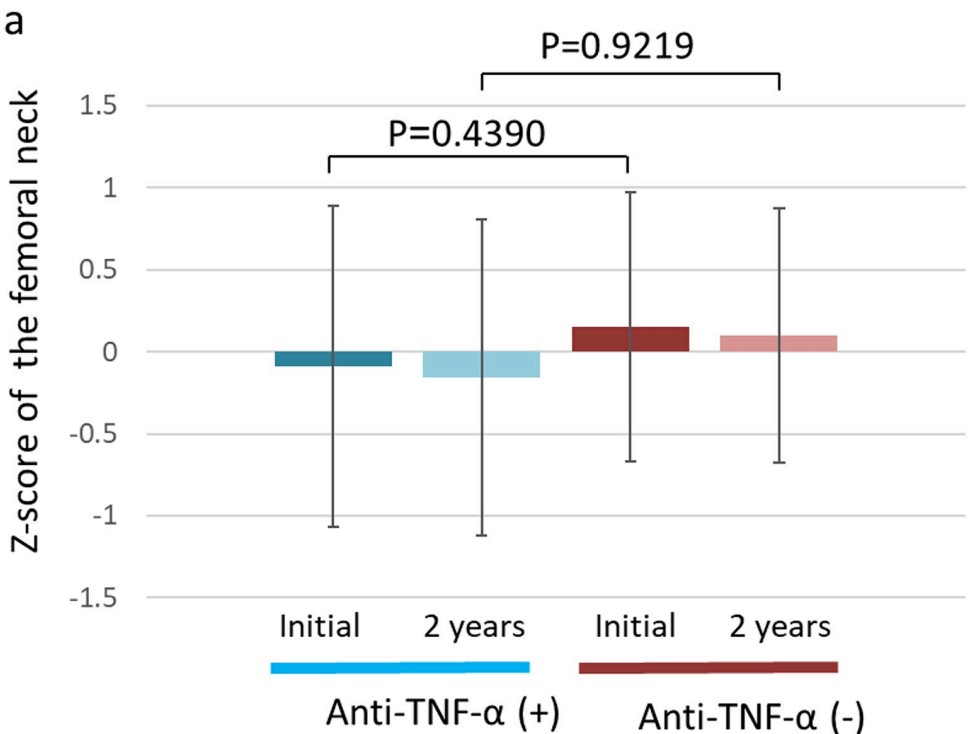

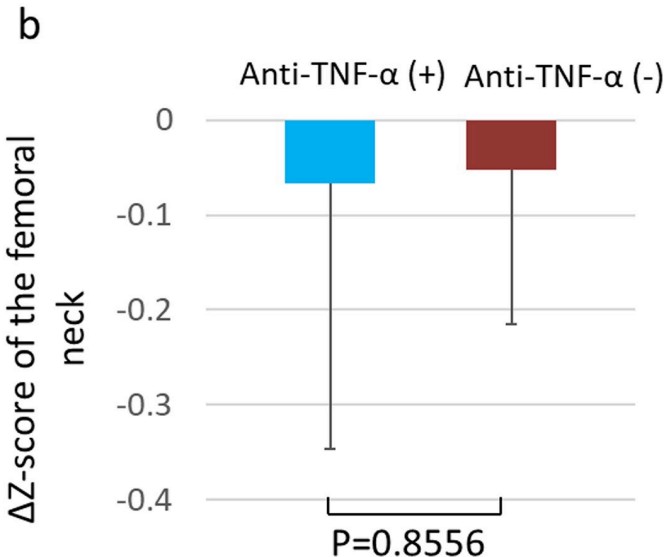

**Fig 4. Change in the Z- and ΔZ-scores of the femoral neck.** (a) Femoral neck Z-scores of the anti-TNF-α administration and non-administration groups at the first measurement and 2 years later. (b) Femoral neck ΔZ-scores of the anti-TNF-α administration and non-administration groups. TNF-α, tumor necrosis factor α.

as anti-TNF-α preparations) on bone metabolism biomarkers. The NTx level significantly decreased after 2 years in CD patients who did not receive the anti-TNF-α therapy; this may be related to the deterioration of bone metabolism. Conversely, the NTx levels did not decrease significantly for 2 years in CD patients who received the anti-TNF-α therapy. Therefore, CD patients receiving anti-TNFα have a relative increase in NTx compared to the normal course of bone metabolism. Although the detailed mechanism is unknown, in this study, it was considered that anti-TNFα administration to CD patients might have contributed to the increase of NTx value. Ryan et al. [44] estimated the NTx level for bone metabolism in CD patients after IFX administration. They reported that although IFX administration did not induce a significant change in the NTx value 4 months after treatment initiation, there was a significant decrease in the NTx value after 16 months, contrary to our study results. This may be due to differences in the disease activity and in the period of anti-TNF-α administration. A further investigation into the same is necessary. Although the detailed mechanism is unclear, it is believed that bone resorption in CD patients is impaired; therefore, these abnormalities have some influence on the decrease in NTx levels. Based on these considerations and the results of this study, we speculate that anti-TNF-α preparations indirectly suppressed the decrease in NTx.

Furthermore, the measurement of NTx as a marker of bone resorption is important in various diseases. Interestingly, we previously reported that serum NTx levels were elevated in CD but not in UC, despite both being an IBD [23]. The difference between CD and UC (or other diseases that cause abnormal bone metabolism) is that CD is associated with a deficiency in the absorption of bone-related nutrients and minerals from the small intestine. Furthermore, previous studies have reported that CD is associated with a higher risk of osteoporosis than UC [24]; the presence of small bowel lesions in CD patients impairs the absorption of vitamin D or calcium.

There are some limitations to this study. First, the number of patients included was small; only 20 and 21 patients were assessed in the two groups. Second, the bone metabolism markers were measured over a relatively short-term period of 2 years; therefore, a long-term evaluation is necessary. Because bone metabolism is expected to change over a long period of time, measurements over a longer period are preferred in this study. Furthermore, NTx has not been compared with serum CTx, an important marker for bone resorption [45].

## Conclusions

Elevated serum NTx could be a useful predictor of femoral BMD loss in patients with CD. Prolonged malnutrition could also be a risk factor for femoral BMD loss. Moreover, anti-TNF-α therapy maintained an elevated serum NTx level, suggesting that treatment with anti-TNF-α may help control the increased bone resorption in CD patients.

## Author Contributions

**Conceptualization:** Natsuki Ishida, Shinya Tani, Yasushi Hamaya, Takahisa Furuta, Ken Sugimoto.

**Data curation:** Natsuki Ishida, Tomohiro Higuchi, Takahiro Miyazu, Satoshi Tamura, Satoshi Suzuki, Mihoko Yamade, Moriya Iwaizumi, Yasushi Hamaya, Satoshi Osawa, Takahisa Furuta.

**Formal analysis:** Natsuki Ishida, Tomohiro Higuchi, Takahiro Miyazu, Satoshi Tamura, Satoshi Suzuki, Mihoko Yamade, Satoshi Osawa, Ken Sugimoto.

**Funding acquisition:** Tomohiro Higuchi, Moriya Iwaizumi, Takahisa Furuta.

**Investigation:** Tomohiro Higuchi, Takahiro Miyazu, Shinya Tani, Mihoko Yamade, Satoshi Osawa, Ken Sugimoto.

**Methodology:** Shinya Tani, Ken Sugimoto.

**Project administration:** Natsuki Ishida, Satoshi Tamura, Takahisa Furuta, Ken Sugimoto.

**Resources:** Tomohiro Higuchi, Takahiro Miyazu, Moriya Iwaizumi, Takahisa Furuta.

**Software:** Tomohiro Higuchi, Shinya Tani, Moriya Iwaizumi, Takahisa Furuta.

**Supervision:** Moriya Iwaizumi, Yasushi Hamaya, Satoshi Osawa, Ken Sugimoto.

**Validation:** Tomohiro Higuchi, Takahiro Miyazu, Satoshi Tamura, Satoshi Suzuki, Shinya Tani, Mihoko Yamade, Moriya Iwaizumi, Satoshi Osawa, Takahisa Furuta.

**Visualization:** Tomohiro Higuchi, Takahiro Miyazu, Satoshi Tamura, Satoshi Suzuki, Shinya Tani, Mihoko Yamade, Satoshi Osawa.

**Writing – original draft:** Natsuki Ishida, Ken Sugimoto.

**Writing – review & editing:** Natsuki Ishida, Yasushi Hamaya, Ken Sugimoto.

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
