## [Decision Letter · Decision Letter 0]

25 Jan 2021

PONE-D-20-31823

Serum N-terminal telopeptide of type I collagen as a biomarker for predicting bone density loss in patients with Crohn disease

PLOS ONE

Dear Dr. Sugimoto,

Thank you for submitting your manuscript to PLOS ONE. After careful consideration, we feel that it has merit but does not fully meet PLOS ONE’s publication criteria as it currently stands. Therefore, we invite you to submit a revised version of the manuscript that addresses the points raised during the review process.

The first part of this manuscript showed that elevated serum NTx could be a useful marker for predicting the loss in femoral bone density in patients with CD, but the data are not strong compared to the existing literature on bone resorption markers. The second part analyzed the relationships between anti-TNFa treatment and serum NTx levels, and the link with bone resorption: the conclusions of this part are not supported by the data.

The submitted study is very similar to previous work from the same authors: Sugimoto et al. Dig Dis Sci 2016. An increased serum N-terminal telopeptide of type 1 collagen, a biochemical marker of increased bone resorption, is associated with infliximab therapy in patients with Crohn’s disease. In this case, PLOS ONE requires that authors provide a sound scientific rationale for the submitted work and clearly reference and discuss the existing literature.

Moreover, the conclusions/interpretations on the relationships between sNTx, osteoporosis and anti-TNFa therapy are not supported by the data, and therefore do not meet this PLOS ONE publication criteria.

Submission cannot be accepted unless these two criteria for publication are met.

We look forward to receiving your revised manuscript.

Kind regards,

Mathilde Body-Malapel

Academic Editor

PLOS ONE

Journal Requirements:

2. Please provide a sample size and power calculation in the Methods, or discuss the reasons for not performing one before study initiation.

3. Please note that PLOS does not permit references to “data not shown.” Authors should provide the relevant data within the manuscript, the Supporting Information files, or in a public repository. If the data are not a core part of the research study being presented, we ask that authors remove any references to these data.

4. Please provide theproduct number and any lot numbers of the immunoassays purchased for your study.

5.We note that you have indicated that data from this study are available upon request. PLOS only allows data to be available upon request if there are legal or ethical restrictions on sharing data publicly. For information on unacceptable data access restrictions, please see http://journals.plos.org/plosone/s/data-availability#loc-unacceptable-data-access-restrictions.

Reviewers' comments:

Reviewer's Responses to Questions

**Comments to the Author**

1. Is the manuscript technically sound, and do the data support the conclusions?

Reviewer #1: Yes

Reviewer #2: No

2. Has the statistical analysis been performed appropriately and rigorously? 

Reviewer #1: Yes

Reviewer #2: I Don't Know

3. Have the authors made all data underlying the findings in their manuscript fully available?

Reviewer #1: Yes

Reviewer #2: No

4. Is the manuscript presented in an intelligible fashion and written in standard English?

Reviewer #1: Yes

Reviewer #2: No

5. Review Comments to the Author

Reviewer #1: Low BMD is considered to be extra-intestinal manifestations of IBD. Managing extra-intestinal complication is a difficult clinical problem and also a timely topic. Although the sample size was small and only 41 patients were included, this study prospectively assessed the association between serum NTx level and bone density loss in CD patients. They found that an elevated serum NTx could be a useful marker for predicting a decrease in the femoral bone mineral density in CD patients. Anti-TNF-α therapy maintained an elevated serum NTx level, suggesting that treatment with anti-TNF-α may help control increased bone resorption in CD patients. The sample size of this study was small, further analyses are recommended.

Minor revision

Personally, using ROC analysis can be added in this study to further confirm the diagnostic and predictive value of serum NTx level and other items.

Reviewer #2: Ishida et al have submitted a manuscript describing a two-year follow-up of a cohort of 41 individuals with Crohn’s Disease, hoping to identify predictors of bone loss (as measured by DXA Z-scores) in this interval. They argue that elevated basal serum NTX values predicted such bone loss, and also try to establish a relationship between treatment with anti-TNFalpha and NTX evolution.

Overall, while the analysis of bone outcomes in persons with CD is well warranted as a research topic, the manuscript is poorly written, some conclusions are not supported by their data, and their interpretation of results is mistaken. Above all, their main finding, that serum NTX at baseline predicted bone loss in 2 years, is absolutely not novel, indeed it is expected – the capacity of bone turnover markers to predict bone loss has been well established in a variety of settings.

Major issues:

1) It is unfortunate that the authors have used serum NTX to analyze bone resorption. An international effort reported by Vasikaran and colleagues in 2011 (Osteoporos Int 22:391–420) has elected serum CTX as the resorption marker of choice, and therefore the bulk of the literature has been focused on this marker, which is also most used internationally in clinical settings. Therefore, the translation of these findings to general practice settings is compromised (would CTX behave in the same fashion – we won’t know). Indeed, because serum NTX determination is seldom used, the authors should describe in more detail its methodology: what is their precision? Intra and inter assay variation? Their main finding is a statistically significant 20 vs 17 difference in s-NTX between groups, but what is the clinical significance of that?

2) Statistical analysis: In tables 2 to 4, a sentence in the footnote claims that “Certain data represent the mean±standard error of mean values”. This is confusing, please be more specific to what each specific point of data (each line) represents.

3) The data shown in Table 4 is not comprehensible. In the text (lines 155-156) and also in the table mention “average of the initial and secondary” measurements, but what is the relevance of showing an average in this case? It would make a lot more sense to show intra-individual variation in biomarker levels between the two points.

4) Lines 187-189 and Figures 1b and 1c: It seems to me that these significant correlations are largely driven by one outlier in each setting. Please perform a statistical analysis excluding outliers to see if these correlations still hold.

5) Lines 198-201 and Figure 2: What is described in the text is the complete opposite of what is shown in Figure 2  Which one is wrong, the text or the group labeling in the figure? Judging by the points that the authors make in the discussion, I believe the figure is right, and that the group that kept high s-NTX levels after two years is the anti-TNFalpha treated group. If that is the case, the authors argue in the discussion (lines 274-275) that “The NTx level significantly decreased after 2 years in CD patients who did not receive the anti-TNF-α therapy; this may be related to the deterioration of bone metabolism” – this concept does not hold. Indeed, the decrease in NTX after two years in this group is a good sign, since the authors themselves have shown that higher s-NTX predicts bone loss! It may well be that the anti-TNF-alpha therapy is stimulating bone resorption on top of the effects of inflammation and malabsorption in people with CD and therefore resulting in more damage to the bone.

6. PLOS authors have the option to publish the peer review history of their article (what does this mean?). If published, this will include your full peer review and any attached files.

Reviewer #1: No

Reviewer #2: No

---

## [Author Response · Author response to Decision Letter 0]

27 Mar 2021

March 22, 2021

Subject: Revised Manuscript (PONE-D-20-31823) “Serum N-terminal telopeptide of type I collagen as a biomarker for predicting bone density loss in patients with Crohn disease”

Dear Prof. Mathilde Body-Malapel:

Thank you for providing us with the opportunity to revise our manuscript (PONE-D-20-31823) as well as for the reviewers’ helpful suggestions and comments. We have addressed the concerns, comments, and questions raised by reviewers 1 and 2 and have presented our detailed point-by-point responses below. Revisions in the manuscript are indicated in red.

We hope that these revisions have improved the quality of our manuscript and have made it suitable for publication. We have uploaded the marked and unmarked copies of our manuscript, as requested.

Response to Editor

The submitted study is very similar to previous work from the same authors: Sugimoto et al. Dig Dis Sci 2016. An increased serum N-terminal telopeptide of type 1 collagen, a biochemical marker of increased bone resorption, is associated with infliximab therapy in patients with Crohn’s disease. In this case, PLOS ONE requires that authors provide a sound scientific rationale for the submitted work and clearly reference and discuss the existing literature.

Authors’ response:

Thank you for your comments. We have previously shown that serum NTx is significantly higher in patients with Crohn's disease than in healthy individuals and patients with ulcerative colitis, and that it tends to be higher, especially in patients treated with infliximab. We showed these results in the published article of Dig Dis Sci in 2016. However, this is a cross-sectional study, only comparing and examining the value at a one-time point in each patient and not performing longitudinal and sequential analysis. In this study, we prospectively examined changes in NTx and bone mineral density over time and longitudinally in the same patient for over 2 years and evaluated the effects of infliximab administration over time. It is clearly different from our previous Dig Dis Sci paper, and we have added this to the Discussion (Page 17-18, Line 258-261)

Moreover, the conclusions/interpretations on the relationships between sNTx, osteoporosis and anti-TNFa therapy are not supported by the data, and therefore do not meet this PLOS ONE publication criteria.

Authors’ response:

Thank you for your comments. As the editor pointed out, there was a lack of data in this paper to support the association between sNTx, osteoporosis, and anti-TNF�. Since reviewers also pointed out this, we added sNTx level ROC analysis to this study and added logistic regression analysis to evaluate the relationship with other factors. As a result, it was shown that NTx ≧ 16.6 nmol BCE/L calculated by ROC analysis is a predictor of Z score decrease in independent CD by univariate and multivariate analysis. We have added this result to the new Table 4 and Results section (Page 10, Line 162-166). Thanks to the suggestions of the editor and reviewers, we are grateful we could add analysis with more clinical significance.

Authors’ response:

Thank you for your comment. We have ensured that our manuscript meets PLOS ONE's style requirements.

2. Please provide a sample size and power calculation in the Methods, or discuss the reasons for not performing one before study initiation.

Authors’ response:

Thank you for your advice. In SPSS analysis, the required sample size was 16 under the condition that the difference in the mean value of NTx between the bone mineral density reduced group and the non-decreased group was "3", the standard deviation was "3", the α error (significance level) was "5% on both sides", and the detection power was "80%". Therefore, in this study, we decided that a sample size of 20 or more in each group would be sufficient. The above comments have been added to the Methods section (Page 8, Line 131-139).

3. Please note that PLOS does not permit references to “data not shown.” Authors should provide the relevant data within the manuscript, the Supporting Information files, or in a public repository. If the data are not a core part of the research study being presented, we ask that authors remove any references to these data.

Authors’ response:

Thank you for your advice. All the descriptions of "data not shown" have been deleted from the text.

4. Please provide theproduct number and any lot numbers of the immunoassays purchased for your study.

Authors’ response:

We thank the editors for the comment. Since we outsourced the samples' measurements to Special Research Laboratory inc., an outside contractor, we do not know the lot number of the measurement samples, including NTx. To avoid any misunderstanding, we have added in the Method section that Special Research Laboratory inc. is an outside contractor (Page 6, Line 98-99).

5.We note that you have indicated that data from this study are available upon request. PLOS only allows data to be available upon request if there are legal or ethical restrictions on sharing data publicly. For information on unacceptable data access restrictions, please see http://journals.plos.org/plosone/s/data-availability#loc-unacceptable-data-access-restrictions. 

Authors’ response:

Thank you for your comment. We understand that data from this study are available upon request.

Response to Reviewers

Reviewer #1: 

Personally, using ROC analysis can be added in this study to further confirm the diagnostic and predictive value of serum NTx level and other items.

Authors’ response:

We are grateful to the reviewers for their appreciation of the progress of this research.

As suggested, we performed NTx level ROC analysis to predict the progression of osteoporosis and created a new Figure 1. As a result of this analysis, the cut-off value of NTx, which predicts the progression of osteoporosis (ΔZ score <0), was 16.6 nmol BCE/L, and the AUC was 0.826 (95% confidence interval 0.694 -0.958). In addition, we added logistic regression analysis to evaluate the relationship with other factors. Univariate and multivariate analysis proved that NTx16.6 or higher is an independent factor with ΔZ score <0. This result has been added to the new Table 4 and Results section (Page 10, Line 162-166). Further, the analysis method is described in the Statistical analysis section (Page 8, Line 131-139)

Thanks to the reviewer’s suggestion, we are grateful for being able to add analyses that will find more clinical significance.

Reviewer #2: 

Overall, while the analysis of bone outcomes in persons with CD is well warranted as a research topic, the manuscript is poorly written, some conclusions are not supported by their data, and their interpretation of results is mistaken. Above all, their main finding, that serum NTX at baseline predicted bone loss in 2 years, is absolutely not novel, indeed it is expected – the capacity of bone turnover markers to predict bone loss has been well established in a variety of settings.

Authors’ response:

Thank you for your advice. It may already be well known that baseline NTx indicates a bone loss in areas other than gastrointestinal disease. However, Crohn's disease has absorption disorder of calcium and vitamin D due to the presence of lesions in the small intestine, and bone metabolism of patients in Crohn’s disease may also be influenced by cytokines, including TNFα. Therefore it has not been sufficiently analyzed whether biomarkers of bone resorption, including NTx, can be directly predictors of bone loss in Crohn’s disease. This time we dared to focus on NTx because we have previously shown in Dig Dis Sci at 2016 that serum NTx is significantly higher in patients with Crohn's disease than in healthy individuals and patients with ulcerative colitis, and that it tends to be higher, especially in patients treated with infliximab. However, this previous study is a cross-sectional study, only a comparative study of one-time point values in each patient and no longitudinal or sequential analysis. Therefore, in this study, we prospectively examined changes in NTx and bone mineral density over time and longitudinally in the same patient for over 2 years and prospectively evaluated the effects of infliximab administration over time. We believe that the new findings we have presented about NTx will be of great clinical benefit to gastroenterologists involved in treating Crohn's disease.

Major issues:

1) It is unfortunate that the authors have used serum NTX to analyze bone resorption. An international effort reported by Vasikaran and colleagues in 2011 (Osteoporos Int 22:391–420) has elected serum CTX as the resorption marker of choice, and therefore the bulk of the literature has been focused on this marker, which is also most used internationally in clinical settings. Therefore, the translation of these findings to general practice settings is compromised (would CTX behave in the same fashion – we won’t know). Indeed, because serum NTX determination is seldom used, the authors should describe in more detail its methodology: what is their precision? Intra and inter assay variation? Their main finding is a statistically significant 20 vs 17 difference in s-NTX between groups, but what is the clinical significance of that?

Authors’ response:

We thank the reviewer for the detailed and accurate suggestions.

As the reviewer pointed out, CTx is the most focused marker of bone resorption. CTx was not analyzed in this study, which was added to the limitation (Page 22, Line 332-333)

Further, as the reviewers pointed out, our study lacked an analysis that considered the usefulness of NTx in clinical practice. In response to this indication, to find further clinical significance of NTx, we attempted to calculate a cut-off value for predicting the progression of osteoporosis of NTx. As a result of ROC analysis, the optimal cut-off value of NTx was 16.6 mmol BCE/L, and the AUC was 0.826 (95% confidence interval 0.694 -0.958). In addition, we performed a logistics regression analysis to assess the risk of the Z score decrease, including NTx, age, gender, disease type, CDAI, surgical history, and use of anti-TNFα preparations. Univariate and multivariate analysis showed that NTx ≥ 16.6 calculated by the ROC analysis described above was an independent predictor of Z score reduction in Crohn’s disease. We added this result to the new Table 4 and Results section (Page 10, Line 162-166). We have also added the methods for performing these analyses to the Statistical analysis section (Page 8, Line 131-139).

We are grateful to the reviewers for further exploration of the clinical significance of NTx.

2) Statistical analysis: In tables 2 to 4, a sentence in the footnote claims that “Certain data represent the mean±standard error of mean values”. This is confusing, please be more specific to what each specific point of data (each line) represents.

Authors’ response:

Thank you for your suggestion to make this manuscript easier to read. In addition, although it was described as a standard error in each table, it was actually a standard deviation, so we corrected it. We have added the mean ± standard deviation of mean values to each particular one in each table.

3) The data shown in Table 4 is not comprehensible. In the text (lines 155-156) and also in the table mention “average of the initial and secondary” measurements, but what is the relevance of showing an average in this case? It would make a lot more sense to show intra-individual variation in biomarker levels between the two points.

Authors’ response:

We thank the reviewer for this suggestion.

As you suggested, we have added a new table to show intra-individual variation in biomarker levels between the two points in the new Table 6 in the revised manuscript. This new Table 6 shows the changes of each biomarker value at first and second measurements in the group of ΔZ score <0 and the group of ΔZ score ≥ 0. Alb showed a significant decrease in the group with ΔZ score <0 (P = 0.02). This result is consistent because hypoalbuminemia in Crohn’s disease is a risk factor for osteoporosis, which has already been described in the discussion section. Interestingly, the NTx value decreased significantly in the group with ΔZ score ≥ 0. In other words, in this study, NTx showed a significant decrease in 2 years in the group of patients whose bone mineral density did not decrease. We added this result in the Results section (Page 11, Line 175-179)

Thanks to the reviewers, we examined the interesting changes in the numbers of each biomarker over the two years. 

4) Lines 187-189 and Figures 1b and 1c: It seems to me that these significant correlations are largely driven by one outlier in each setting. Please perform a statistical analysis excluding outliers to see if these correlations still hold.

Authors’ response:

We appreciate the reviewers' suggestions.

We analyzed the correlations of the correlation diagrams in Figures 1c and 1b, excluding outliers, and showed significant correlations (Data not shown). We presented the original correlation diagram without changing it.

5) Lines 198-201 and Figure 2: What is described in the text is the complete opposite of what is shown in Figure 2  Which one is wrong, the text or the group labeling in the figure? Judging by the points that the authors make in the discussion, I believe the figure is right, and that the group that kept high s-NTX levels after two years is the anti-TNFalpha treated group. If that is the case, the authors argue in the discussion (lines 274-275) that “The NTx level significantly decreased after 2 years in CD patients who did not receive the anti-TNF-α therapy; this may be related to the deterioration of bone metabolism” – this concept does not hold. Indeed, the decrease in NTX after two years in this group is a good sign, since the authors themselves have shown that higher s-NTX predicts bone loss! It may well be that the anti-TNF-alpha therapy is stimulating bone resorption on top of the effects of inflammation and malabsorption in people with CD and therefore resulting in more damage to the bone.

Authors’ response:

We are most grateful for the reviewers' suggestions.

As the reviewer pointed out, the description in the main text is incorrect. “In the anti-TNF-αadministration group, the NTx value did not differ significantly between the first and second measurements (P = 0.8735) (Fig. 3). However, In the anti-TNF-α non-administration group, the NTx value was significantly lower at the second measurement than at the first measurement (P = 0.0328)” is correct. As the reviewer suggested, this part has been corrected in the text (Page 16, Line 226-230). We deeply apologize for our careless mistakes. In addition, as the reviewer pointed out, the second half of the sentence was stated in the Discussion section, "The NTx level significantly decreased after 2 years in CD patients who did not receive the anti-TNF-α therapy; this may be related to the deterioration of bone degradation." did not match the explanation of Fig. 2 before the revision. As shown by the changes in NTx between initial and 2 years added in new Table 6, NTx decreased significantly in 2 years in the group of patients with ΔZ score ≥ 0 whose bone metabolism was normal. As mentioned above, the NTx value of the anti-TNF-α non-administration group was significantly decreased in Fig. 2, showing the same course as the NTx of the patients with normal bone metabolism (ΔZ score ≥ 0 group). Nevertheless, NTx in the anti-TNF-α administration group did not show a significant change, and it is considered that NTx increased relatively when considering changes in NTx levels in normal bone metabolism. From the above, it was found that anti-TNF-α administration affects bone resorption in CD patients. Based on the above results, the discussion section has been revised (Page 20, Lines 310-313)

We are most grateful to the reviewers' suggestions for providing a more consistent and corroborative interpretation.

---

## [Decision Letter · Decision Letter 1]

12 Apr 2021

Serum N-terminal telopeptide of type I collagen as a biomarker for predicting bone density loss in patients with Crohn disease

PONE-D-20-31823R1

Dear Dr. Sugimoto,

We’re pleased to inform you that your manuscript has been judged scientifically suitable for publication and will be formally accepted for publication once it meets all outstanding technical requirements.

Kind regards,

Mathilde Body-Malapel

Academic Editor

PLOS ONE

Additional Editor Comments (optional):

Reviewers' comments:

Reviewer's Responses to Questions

**Comments to the Author**

1. If the authors have adequately addressed your comments raised in a previous round of review and you feel that this manuscript is now acceptable for publication, you may indicate that here to bypass the “Comments to the Author” section, enter your conflict of interest statement in the “Confidential to Editor” section, and submit your "Accept" recommendation.

Reviewer #1: All comments have been addressed

2. Is the manuscript technically sound, and do the data support the conclusions?

Reviewer #1: Yes

3. Has the statistical analysis been performed appropriately and rigorously? 

Reviewer #1: Yes

4. Have the authors made all data underlying the findings in their manuscript fully available?

Reviewer #1: Yes

5. Is the manuscript presented in an intelligible fashion and written in standard English?

Reviewer #1: Yes

6. Review Comments to the Author

Reviewer #1: This revised research added the ROC analysis and found the AUC of NTx, which predicts the progression of osteoporosis (ΔZ score <0), was 0.826. That is to say, NTx showed good diagnostic and predictive value. Besides, received comments also answered problems proposed by the other reviewer. Personally, this revised manuscript could be accepted.

7. PLOS authors have the option to publish the peer review history of their article (what does this mean?). If published, this will include your full peer review and any attached files.

Reviewer #1: No

---

## [Editor Report · Acceptance letter]

16 Apr 2021

PONE-D-20-31823R1 

Serum N-terminal telopeptide of type I collagen as a biomarker for predicting bone density loss in patients with Crohn disease 

Dear Dr. Sugimoto:

I'm pleased to inform you that your manuscript has been deemed suitable for publication in PLOS ONE. Congratulations! Your manuscript is now with our production department. 

Kind regards, 

on behalf of

Dr. Mathilde Body-Malapel 

Academic Editor

PLOS ONE